# Selecting causal brain features with a single conditional independence test per feature

**Atalanti A. Mastakouri**
Empirical Inference Department
Max Planck Institute
for Intelligent Systems
Tübingen, 72076
amastakouri@tue.mpg.de

**Bernhard Schölkopf**
Empirical Inference Department
Max Planck Institute
for Intelligent Systems
Tübingen, 72076
bs@tue.mpg.de

**Dominik Janzing**
Amazon Research
Tübingen, 72076
janzind@amazon.com

## Abstract

We propose a constraint-based causal feature selection method for identifying causes of a given target variable, selecting from a set of candidate variables, while there can also be hidden variables acting as common causes with the target. We prove that if we observe a cause for each candidate cause, then a single conditional independence test with one conditioning variable is sufficient to decide whether a candidate associated with the target is indeed causing it. We thus improve upon existing methods by significantly simplifying statistical testing and requiring a weaker version of causal faithfulness. Our main assumption is inspired by neuroscience paradigms where the activity of a single neuron is considered to be also caused by its own previous state. We demonstrate successful application of our method to simulated, as well as encephalographic data of twenty-one participants, recorded in Max Planck Institute for intelligent Systems. The detected causes of motor performance are in accordance with the latest consensus about the neurophysiological pathways, and can provide new insights into personalised brain stimulation.

## 1 Introduction

Conditional independence (CI) relations have been an important tool in the field of computational statistics [1, 2] and play a significant role in causal inference [3]. However, causal inference through conditional independencies in real datasets is a challenging task, since testing them is a hard task [1], particularly when the number of conditioning variables is large. PC [4], FCI [4] and CPC [5] are three of the most prominent CI based causal discovery methods. To recover the underlying graph from the data they require some assumptions, which, however, are often violated. These include the *causal Markov condition, faithfulness* and, in addition for PC method, also *causal sufficiency*, i.e., the assumption that all common causes of observed nodes are observed. Although FCI algorithm [4] does not assume that, it becomes unreliable because it requires many statistical tests if the connections between the features are not sparse. Furthermore, faithfulness is a rather problematic assumption, as typical parameter values in causal models with many variables yield distributions that are close to being unfaithful [6].

The field of non-invasive neuroimaging, such as Electroencephalography (EEG), is one characteristic case where the discovery of causal features is needed. There, the activity of billions of neurons is recorded as noisy mixtures of activity reaching through several layers of cortex, skull and skin and hence causal sufficiency cannot be assumed. Furthermore, the dimensionality of the data is large, often comparable to the sample size. In such datasets, the need for causal inference often arises, in order to be able to differentiate a set of causal brain features from a large number of simple correlations between the brain activity and the observed behavioral response [7, 8, 9].

Our motivation emanates from the field of non-invasive brain stimulation (NIBS); a novel treatment tool that aims, among others, at the rehabilitation of motor functions, for patients with motor disabilities. One fundamental problem is the lack of exact knowledge of the mechanism that entrains the ongoing brain oscillations during the stimulation [10, 11]. Subsequently, the selection of the frequency, intensity and exact location of the stimulation is made based on collected observations, instead of being derived from the individual's brain activity. For instance, stimulation at $\gamma$-range frequencies (70Hz) has been proposed to facilitate movement [12, 13], while frequencies in $\beta$-range have been reported to inhibit it [14, 15, 16]. However, particularly in motor tasks, similar stimulation parameters have been reported to result in very heterogeneous responses across subjects, that span from positive to negative [17, 18]. It has been argued [19] that the reason for this discrepancy of responses to NIBS originates from the limited or extensive variability of each brain's activity during movement, and hence personalized stimulation parameters are required to ensure positive response. Better understanding of the motor cortex activation of each individual could contribute to the identification of such individual parameters.

Here we present a constraint-based causal feature selection method for identifying causes of a target variable, in set-ups where one cause for each candidate cause is known. We prove that the detection of causes remains unaffected by common causes, regardless of whether they are observed or unobserved. Our method restricts the identification of a cause to one targeted conditional independence test per candidate, with only one conditioning variable. This restriction simplifies the required statistical testing by both limiting the problem of possible faithfulness violation and by scaling with linear complexity with the number of features. As a first step, we apply our causal methodology on simulated data, which leads to very low percentages of false positives due to statistical error. While the application of our proposed algorithm is not restricted to brain datasets, we apply it also on EEG data that we recorded from twenty-one healthy subjects during a motor task, with the aim of detecting causal brain features of motor performance for each individual. We give evidence of different detected causal brain features according to the motor performance, which are in accordance with the state-of-the-art research in the field of neuroscience. We then discuss how this could be used to identify personalized stimulation parameters for rehabilitation.

## 2 Methods

### 2.1 Definitions and notations

We briefly present some fundamental definitions in Causal Bayesian Networks [20], which we will use to present our methodology and prove our theorem below. For a thorough study see [20, 21, 22]. The notions of *faithfulness* and of *causal Markov condition* are fundamental to be able to relate the distributions of the variables of interest to properties of a causal graph. Markov condition enables us to read off *independences* from the graph structure, while faithfulness allows us to infer *dependences* from the graph [21]. In other words, a distribution $P$ is faithful to a directed acyclic graph (DAG) $G$ if no conditional independence relations other than the ones entailed by the Markov property are present. Another important notion for causal discovery is the *confounding path* between two variables. Here, we define a confounder variable as a variable (observed or unobserved) that is a common ancestor of two other variables. In Appendix A.1 we give a list of the exact definitions for the terms that we use (*d-separation, Ancestor, Descendant, Causal Markov Condition, Faithfulness, Confounder*).

From now on, we are going to use that following notation, to describe our method:

- $\dashrightarrow$: denotes a directed path with observed variables or a direct link.
- $\rightarrow$: denotes a direct link.

We briefly introduce the environment of our methodology. The problem of selecting causal features is inspired from brain datasets, where the causal candidates are brain features $i$; i.e. activity in different brain regions and in different brain frequencies, and the target variable $R$ is a behavioural response we measure on the subject. We also consider that each candidate variable has an observed previous $P^i$ and a current $M^i$ state of the brain feature $i$. The variables' names read as "Plan", "Move" and "Response" respectively. An example of such a structure is given in Figure 1, where for example (brain) feature $M^1$ has an ancestor $P^1$ and is a cause of $R$, while feature $M^2$ is not causing $R$ but connects with a confounding path that includes $M^1$. Without knowing the structure, our theorem

is able to differentiate the true causes ($M^1$) from the ones that are dependent to the target due to confounding paths ($M^2$).

## 2.2 Formal problem description

Given the random variables $P^i$, $M^i$ $i = 1, 2...n$ and $R$, we assume the class of DAGs in which there can be instantaneous acyclic effects between $P^i$ variables, between $M^i$ variables, as well as forward effects between $P^i$, $M^i$ and $R$. In Section 2.3 we explain how the assumptions described below are commonly met in datasets where candidate causes can be measured in two time stamps, and hence a causal path from the previous to the current state can be assumed. Such a case is a brain set-up. Below we present the necessary assumptions for our theorem.

**Assumption** (1). *Causal Markov condition*

**Assumption** (2). *Faithfulness*

**Assumption** (3). $P \not\dashleftarrow M \not\dashleftarrow R$ *: In the class of DAGs we target here, variable $R$ is measured after $M$, which is measured after $P$ (there can be no backwards arrows in time).*

**Assumption** (4). $P^i \dashrightarrow M^i$ *exists: Variables $P^i$ and $M^i$ represent two consecutive states of the same brain feature $i$. We assume that the state $P$ is always a cause of state $M$ for the same feature $i$.*

**Assumption** (5). $(R, M^i, P^i)$ *are independently drawn from some distribution (i.i.d)*

**Theorem.** *Given the variables $P^i$, $M^i$ $i = 1, 2...n$ and $R$, and assuming 1-5, if $M^i \not\perp\!\!\!\perp R$ (1) and $P^i \perp\!\!\!\perp R \mid M^i$ (2) then $M^i \dashrightarrow R$.*

*Proof.* We prove the claim by contradiction. Assume $1 - 5$ and that $M^i$ and $R$ are dependent (cond. 1), but there is no directed path from $M^i$ to $R$. Then there is a confounding path $p_1 := M^i \dashleftarrow C \dashrightarrow R$ with some common cause $C$ (hidden or observed). Now consider some path $p_2 := P^i \dashrightarrow M^i$ (it exists due to Assumption 4). If $p_1$ and $p_2$ have only $M^i$ in common, $M^i$ is a collider and thus $P^i$ and $R$ are not d-separated by $M^i$. If $p_1$ and $p_2$ share more nodes, assume first they have $P^i$ in common, that is, $P^i$ lies on $p_1$. Then $P^i$ and $R$ are not d-separated by $M^i$ because the sub-path of $p_1$ connecting $P^i$ and $R$ does not contain $M^i$ and $p_1$ is collider-free. Assume now that $P^i$ does not lie on $p_1$, and $p_1$ and $p_2$ share some node $X$ other than $M^i$ and $P^i$. Then either (i) $X = C$, or (ii) $X$ is a node between $C$ and $R$, or (iii) $X$ is a node between $C$ and $M^i$. For (i) and (ii), we have a directed path from $P^i$ to $R$ (that does not contain $M^i$). In case (iii), $X$ is a collider and $M^i$ a descendent of this collider, hence $M^i$ unblocks the path from $P^i$ to $R$. In all three cases, $M^i$ does not d-separate $P^i$ and $R$, which contradicts $P^i \perp\!\!\!\perp R \mid M^i$ (cond. 2) due to faithfulness. Hence there must be a directed path $M^i \dashrightarrow R$.

$\square$

Note that our algorithm requires only one CI test for each node. Therefore, it speeds up the causal feature selection as it scales linearly with the number of nodes in the graph; hence its complexity is $\mathcal{O}(n)$. The Matlab code can be found in https://gitlab.tuebingen.mpg.de/amastakouri/singleCICausalFeatureSelection.git

---

**Algorithm 1:** Find causes of $R$

---

**Input:** $P^i, M^i, R, \forall i = 1, ..., n.$
**Output:** $CausesR$
**for** $i \leftarrow 1$ **to** $n$ **do**
    $pvalue1 \leftarrow ind\_test(M^i, R)$
    **if** $pvalue1 < threshold1$ **then**
        $pvalue2 \leftarrow cond\_ind\_test(P^i, R, M^i)$
        **if** $pvalue2 > threshold2$ **then**
            $CausesR \leftarrow [CausesR, M^i]$
        **end**
    **end**
**end**

---

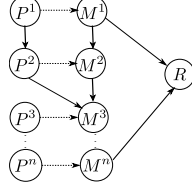

**Figure 1:** A possible DAG that includes the random variables $P^i$, $M^i$ $i = 1, 2...n$ and $R$, assuming 1-5. Each candidate causal feature $M^i$ have a cause $P^i$ and may have other acyclic edges with the other candidates, and some features $M^i$ cause target $R$.

Note that Theorem 2.2 provides sufficient but not necessary conditions for $M^i$ to be a cause of $R$. In other words, not all causes of $R$ may be identified. Note that our assumptions do not include causal sufficiency. So even in the case of unobserved common causes, if the conditions described in 2.2 are met, then we know that the dependency between the $M^i$ and $R$ is due to a directed path and not due to a confounder variable. (We should check that the relationship between $M^i$ and $P^i$ is not too deterministic. Obviously this would amount to the conditional independence $P^i \perp\!\!\!\perp R \mid M^i$ even in the presence of counfounding. This violation of faithfulness could happen if $M^i$ and $P^i$ are too close in time.)

To corroborate our statement, we further explore the following case: In real datasets where the different samples have been measured in different timestamps (such as our EEG experiment), even if the interval between measurements is large, we should not consider the samples as i.i.d. data (Assump. 5). There is, however, a heuristic argument suggesting that under Assumptions 1-4, our method is robust with respect to the i.i.d. violation. To this end, we model the time dependence formally by a hidden time variable $T$. We examine the conditions of Theorem 2.2 exhaustively on each of the different cases that $T$ can affect the variables of the DAG; i.e. one variable at a time, two and then all of them (see Fig. 6 in suppl.). In case that $M^i \dashrightarrow R$ exists, condition (2) is violated in the graphs (a), (c) and (d), since $P^i \not\perp\!\!\!\perp R \mid M^i$. In all the other graphs of Fig. 6 (suppl.), the cause $M^i$ is correctly identified. If $M^i \not\dashrightarrow R$ (Fig. 7 in suppl.), then graphs (a), (c) and (d) comply with condition (1) of Theorem 2.2, but violate condition (2), correctly rejecting the variable $M^i$. For the rest of the graphs, the variables already violate condition (1), and are thus rejected. Therefore, if the hidden variable $T$ is present, some causal variables may be rejected but no non-causal variable is falsely accepted. This is desired in applications where false positives are harmful compared to false negatives.

## 2.3 Experimental part

We apply our method on simulated data and on EEG data that we recorded from twenty-one healthy participants. All EEG experiments and recordings were performed in Max Planck Institute for Intelligent Systems under the ethics approval of the Committee of the Eberhard Karls University of Tübingen. Informed consent was obtained by all participants, prior to their participation to the study. For the implementation, to make sure that in practice Assumption 4 in the data is not violated, we check the dependence between $P^i$ and $M^i$ for the same $i$, with an independence test, and in case it is not significant we reject the candidate without further checking. Both for the simulated graphs described below and for the EEG data, we calculate the independencies using the HSIC test [23] and the conditional independencies using the conditional independence HSIC test from [24, 25], with Gaussian kernel and the usual heuristic bandwidth used in [23]. Therefore, our algorithm also checks for non-linear relationships between the variables. For the statistical testing we examine the null hypothesis $H0_1 : M^i \perp\!\!\!\perp R$ and consider to have rejected the null hypothesis (hence consider to have found $M^i$ and $R$ to be dependent) if $p < \alpha_D = 0.05$. Then, we examine the null hypothesis $H0_2 : P^i \perp\!\!\!\perp R \mid M^i$ and accept it (hence the conditional independence) if $p > \alpha_{CI} = 0.25$ (usual values for accepting CI in EEG datasets include thresholds above 0.25 [8]).

### 2.3.1 Simulated graphs

Given the variables $P^i$, $M^i$ $i = 1, 2...n$ and $R$ as described in 2.2, and assuming 1-5, we build simulations of possible DAGs and apply our Theorem 2.2. Simulations were run on a 12-CPU computer using the parallel toolbox of Matlab.

**Construction of simulated graphs:** We sample the noise terms of $P$, $M$ and $R$ variables from a Gaussian distribution with variance randomly sampled from a uniform distribution. We then define the adjacency matrix of the subgraph that consists of all $P^i$ variables as an $n \times n$ matrix $A_P$, whose elements are independently drawn from a Bernoulli($p$) distribution, denoting the existence of an edge between the different $P^i$ variables, forbidding any self-cycles ($a_{P_{i=j}} = 0$). We update the $P^i$ values by adding a function $f_1$ of each parent $P^j$ variable: $P^i = P^i + \sum_{j=1}^{k_P} f_1(P^j_{a_{P_{ij}}==1})$, for the $k_P$ parent $P^j$ variables of $P^i$. As a second step, we define the adjacency matrix of the subgraph that consists of all $P^i$ and $M^i$ variables as a $n \times n$ matrix $A_{\text{PM}}$, whose elements are values independently drawn from a Bernoulli($p$), denoting the existence of an edge between the different $P^i$ and $M^i$ variables, making sure that for $i = j$ the edge exists ($a_{\text{PM}_{i=j}} = 1$). We update the $M^i$ values by adding a function $f_2$ of each parent $P^j$ variable: $M^i = M^i + \sum_{j=1}^{k_{\text{PM}}} f_2(P^j_{a_{\text{PM}_{ij}}==1})$, for the $k_{\text{PM}}$ parent $P^j$ variables of $M^i$.

To avoid creation of cycles, we only generate the following types of arrows: (1) $P^i \rightarrow M^j$ for $i \leq j$, (2) $P^i \rightarrow P^j$ for $i < j$, (3) $M^i \rightarrow M^j$ for $i < j$, (4) $P^i \rightarrow R$ and (5) $M^i \rightarrow R$. As a third step, we create the adjacency matrix of the subgraph that consists of all $M^i$ variables as a $n \times n$ matrix $A_M$, whose elements are values independently drawn from a Bernoulli($p$), denoting the existence of an edge between the different $M^i$ variables, forbidding any self-cycles ($a_{M_{i=j}} = 0$). We update the $M^i$ values by adding a function $f_3$ of each parent $M^j$ variable: $M^i = M^i + \sum_{j=1}^{k_M} f_3(M^j_{a_{M_{ij}}==1})$, for the $k_M$ parent $M^j$ variables of $M^i$. Finally, we create the vectors $A_{\text{MR}}$ and $A_{\text{PR}}$ with $n$ elements independently drawn from a Bernoulli($p$), denoting the existence of an edge from $M$ to $R$ and from $P$ to $R$. We update the $R$ values by adding a function $f_4$ of each $M^i$ that is a parent and a function $f_5$ of each $P^i$ that is a parent: $R = R + \sum_{i=1}^{k_{\text{MR}}} f_4(M^i_{a_{\text{MR}_i}==1}) + \sum_{i=1}^{k_{\text{PR}}} f_5(P^i_{a_{\text{PR}_i}==1})$, for the $k_{\text{MR}}$ parent variables $M^i$ and the $k_{\text{PR}}$ parent variables $P^i$ of $R$. We sample the coefficients for the five linear functions $f_1, f_2, f_3, f_4, f_5$ from a Gaussian distribution. We examine the statistical performance of our algorithm for different number of nodes $n$ for the $P$ and $M$ variables, sparsity of edges and different number of samples. For each combination, we examine 20 random graphs and report the percentage of the false positives and false negatives, calculated on the number $n$ of features $i$.

**Comparison with Markov Blanket methods and Lasso:** Lasso or Markov Blanket (MB) discovery methods require causal sufficiency, let alone curse of dimensionality. Furthermore, with high dimensional data, any algorithm using CI tests has to condition on large variable sets, in which case CI testing is hard [1] and cannot be trusted unless sample sizes are huge. Finally, even if causal sufficiency were to hold, the known MB detection algorithms and Lasso do not *detect* variables but rank them, and gradually evaluate the prediction accuracy by including more variables, according to the ranked order the algorithm returned. This requires a heuristic *hyperparameter* to define what is the right acceptable number of variables to be included in the MB, which affects the false positive and the false negative rates. For completeness, however, we provide comparison results of our method against the following three available algorithms (average for 10 random graphs): HSIC Lasso [26], Backwards elimination with HSIC, and Forward selection with HSIC for MB discovery [27]. We present the most optimistic for the other algorithms case, that of large sample size (800) and two cases of small (20) and large graphs (125 nodes), for sparse (0.2) and dense (0.5, more true causes) edges. We report the % of false positives and false negatives in the number of variables.

### 2.3.2 Identifying brain causes of motor performance from EEG data

Our motivation behind the development of this method was to identify causal brain features of upper limb movement from brain activity during a motor task, which could help to identify targets of personalised non-invasive brain stimulation. Here, we apply our method to EEG data (no brain stimulation applied), independently for each subject. Our causal candidate variables are bandpower in different frequency bands and in different electrode locations.

We recorded twenty-one healthy participants with high density EEG (128 electrodes, Brain Products), during a motor task. Our paradigm consisted of 150 trials. During each trial, a new target appeared on a randomized location on a screen in front of the subject. After a planning period of $2.5 - 4\ s$, subjects had been instructed to move their right arm to reach the target within $10\ s$. Subject's arm was being tracked in real time with four infrared cameras (PhaseSpace) and was represented on the screen as a sphere which they could control.

Each trial $k$ consisted of a planning phase $(p_k^i)$ followed by a moving phase $(m_k^i)$. Trials in which the subject did not reach the target within the $10s$-window are excluded from the analysis. As an input to our causal discovery algorithm, we examine the bandpower of four brain frequency bands ($\alpha : (8 - 12)$Hz, $\beta : (12 - 25)$Hz, low-$\gamma : (25 - 45)$Hz and $\gamma : (60 - 80)$Hz) and thirty-eight electrodes over the left and right primary motor cortices, the supplementary motor areas and the central sulcus. That results in $n = 4 \times 38 = 142$ features. We calculate each feature $i$ as the log-bandpower during a window of $1$ $s$ in the end of the planning phase ($P^i$) and in the beginning of the moving phase ($M^i$) for the aforementioned four canonical brain frequency bands (larger interval between the period of $P^i$ and $M^i$ calculation was also examined, which led to less detected causes). Finally, we quantify the response $R$ as the natural logarithm of duration of the reaching movement in seconds (see Figure 1). Each sample of the random variables $P^i$, $M^i$ and $R$ is one experimental trial. We assume the interval between the trials is wide enough to consider them i.i.d. (Assump. 5). In Section 2.2 we examine the violation of this assumption. Assumption 3 and 4 arise in a natural way from an EEG set-up: There is a time ordering between the brain states $P^i$, $M^i$ and $R$; that is why the measured response $R$ cannot affect the preceding brain state (Assump. 3). In addition, we assume that the previous state of brain feature $i$ ($P^i$) is a cause of its current brain state $M^i$ (Assump. 4).

**Preprocessing of EEG data:** Before the bandpower calculation, to attenuate non-cortical artifacts in the EEG data we followed a standardized procedure often applied in this field [28, 29]. We filtered the EEG signal with a Butterworth $3$ $Hz$ high-pass filter, performed common average referene filtering on all electrodes, and then performed SOBI [30] Independent Component Analysis (ICA) followed by manual rejection of non-cortical sources [31], which then we re-projected on the raw signal.

## 3 Results

### 3.1 Simulated data

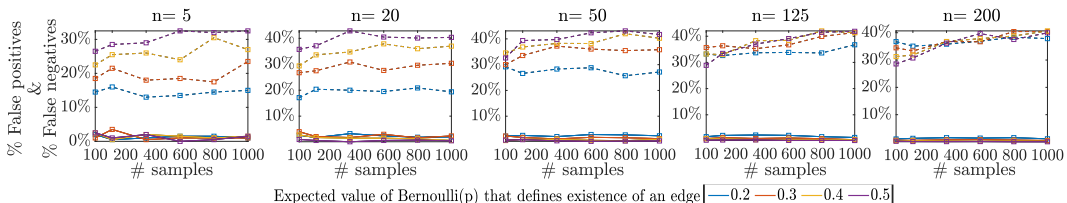

Expected value of Bernoulli(p) that defines existence of an edge — 0.2 — 0.3 — 0.4 — 0.5

**Figure 2:** Percentage of false positives (FP) and false negatives (FN) of detected causes, calculated on the number of features $n$, for twenty random simulated graphs, for different sparsity of edges, number of samples and number of features $i$. Solid lines: FP. Dashed lines: FN. FN increase with the number of nodes. FP due to statistical error remain very low regardless of the number of nodes.

Figure 2 depicts the percentage of false positives and false negatives over twenty random graphs, for each combination of number of $M^i$ nodes $n$, samples and sparsity of edges. As shown in detail in Fig. 8 of suppl., the false positives occurring due to statistical error in the computation of the dependencies and conditional independences are very few, with a tendency to reduce with more samples. Clearly, the probability of false positives increases with the number of nodes. The number of false negatives (Fig. 9 suppl.) appears inflated because we consider as true causes both the direct and the indirect ones; so in case only the direct cause is correctly identified, then its ancestors which are indirect causes will be counted as false negatives. That is why the number of false negatives increases with the number of features $n$ and the density of the graph.

**Comparison with Markov Blanket methods and Lasso:** In the simulated data, in sparse large graphs Forward Selection gave more false positives (table 1). Lasso and Forward Selection gave more false positives in small sparse and dense graphs. Backward Elimination performed worse in small sparse graphs. Overall, our method managed to keep the false positive rate very low ($\sim 2.1\%$) for all dense/sparse, small/large graphs, while other algorithms' performance varied with the case. Optimal parameters based on the true number of causes was selected for Lasso. Backward Elimination and Forward Selection computations took significantly long. Furthermore, we stress that in these simulations no hidden variables exist, which is an extra advantage for the compared algorithms.

**Table 1:** Comparison of false positive and false negative rates calculated in 10 random simulated graphs, among our method and Forward Selection, Backward elimination for markov blanket detection and HSIC Lasso.

| (nodes, sparse) | FP(%) | FN(%) | FP(%) | FN(%) | FP(%) | FN(%) | FP(%) | FN(%) |
|---|---|---|---|---|---|---|---|---|
| | **Our method** | | **Hsic Lasso** | | **BE hsic** | | **FS hsic** | |
| (20,.2) | **3.5** | 31.5 | 9.5 | 22.5 | 11 | 23 | 6 | 25 |
| (20,.5) | **2** | 80 | 5.5 | 47.5 | 1.5 | 79 | 7.5 | 26 |
| (125,.2) | **2.9** | 70.3 | 1.1 | 77.4 | 1.4 | 77.9 | 7.8 | 47.4 |
| (125,.5) | **0** | 80.8 | 0 | 84.8 | 0 | 97.6 | 1.1 | 14.5 |

## 3.2 Electroencephalographic data

Our findings are consistent across all subjects and divided in three categories that couple detected causes with subjects' performance: 1. $\gamma$-power is detected when subjects improve their performance, 2. $\beta$-power is detected when subjects worsen or do not improve their performance, and finally 3. $\alpha$-power is detected in the ipsilateral hemisphere. The three groups are discussed in Section 4.

**Table 2:** Detected causes for six representative subjects; two subjects for each of the three categories of detected causes: 1. $\beta$-range detected causal electrodes for inhibition of performance (AB and DC), 2.$\gamma$-range detected causal electrodes for improvement of performance (KK and II), 3. $\alpha$-range detected causal electrodes over ipsilateral hemisphere (HH and JJ).

| Subject | Alpha | Beta | Low Gamma | Gamma | Above Group Average | Performance |
|---|---|---|---|---|---|---|
| AB | - | FC2, CCP1h, CPP1h, CP6 | - | - | False | Full inhibition |
| DC | FCC5h | CPP2h, CP5, CPz | C2, CCP2h | FC5, CCP2h | False | Inhibited but then improved |
| KK | C6, CP2 | CCP4h, CCP3h, FCC3h, FCC5h, FCC6h, CP3 | FCC2h, CP3, CP1, CP2 | FC5, CCP4h, C6, CCP6h, FC6, FCC3h | False | Full improvement |
| II | - | - | - | FC2, FC4 | False | Full improvement |
| HH | FC2, FCz | - | - | - | False | Improvement but then inhibited |
| JJ | FC4, FC6 | - | - | FCC5h, CP1 | False | Full improvement |

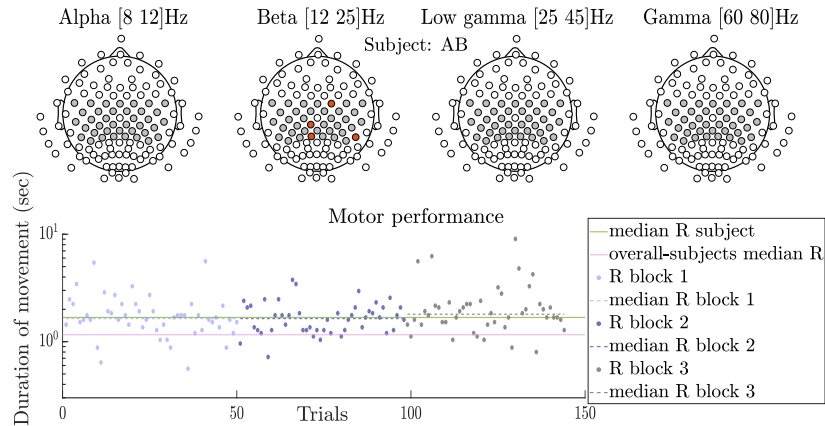

**Figure 3:** Electrodes over contralateral motor and parietal cortex in the $\beta$-range (colored red, 2nd plot) are detected as causal features from our algorithm, for subject AB, who worsened her movement duration during the reaching trials. Findings are in line with literature about the inhibitory role of beta power. Grey color depicts the motor channels we examine. The y-axis is in logarithmic scale.

We applied our method on the preprocessed EEG data described in 2.3.2, individually for each subject. In total, our algorithm identified causes in seventeen out of twenty-one subjects. Due to lack of space, here we present results for six representative subjects in Table 2 and visualisation for two subjects in Figures 3 and 4. Subject AB (Fig. 3) and DC in Table 2 are two representative subjects who worsened or did not improve their movement duration throughout the sequence of reaching trials (larger durations for completing the trial). Subject AB performed on average (green line) worse than the median performance of all subjects (pink line). Our algorithm detected causes over motor channels in the $\beta$-range (2nd head-plot), as well as a few in gamma range (for subject DC in table).

Subjects KK and II (Fig. 4) improved their performance, decreasing the duration of their reaching movements throughout the trials. Our algorithm detected causes over motor channels in the $\gamma$-range (4th head-plot), for both subjects. Finally, HH and JJ are two representative subjects for whom our algorithm detected causes over ipsilateral motor channels in the $\alpha$-range. Results for each subject are presented and explained based on their performance in Section A.5 (suppl.).

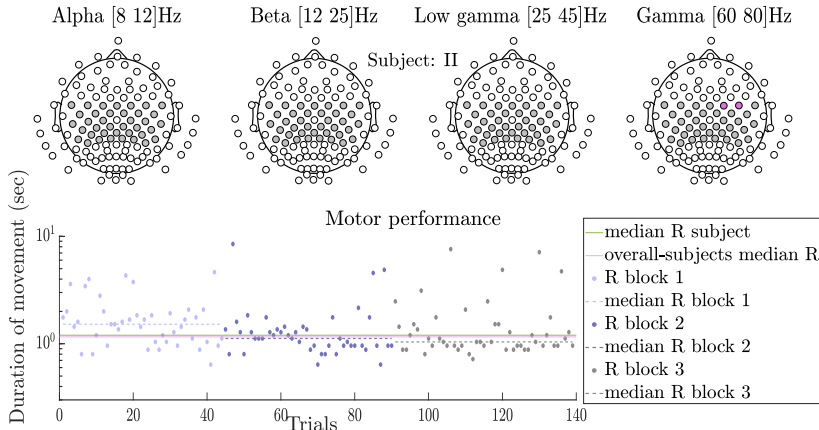

**Figure 4:** Electrodes over motor cortex in the $\gamma$-range (colored pink, 4th plot) are detected as causal features from our algorithm, for subject II, who improved her movement duration over the trials. Findings are in line with literature about the prokinetic role of gamma power. Grey color depicts the motor channels we examine. The y-axis is in logarithmic scale.

## 4   Discussion

**Improvements upon previous methods.** To the best of our knowledge, this is the first constraint based algorithm that scales linearly with the number of candidate features. Previous methods based on CI tests grow exponentially in time with the number of variables, (if sparse data then they grow polynomially), as they require more than one CI test per variable. Therefore, we greatly reduce the computational complexity. Moreover, our algorithm builds on tests that condition on only one variable each. With this improvement, the statistical strength of our inference is superior compared to algorithms where there is more than one conditioning variable. Furthermore, due to this improvement we require a weaker notion of faithfulness [6], as we only assume one triplet of variables per candidate cause. As a third point, our method does not assume causal sufficiency - a common assumption which is, however, often violated in real datasets. Finally, although originally for completeness we assume i.i.d. samples, we prove in the suppl. that our method is robust against false positives when the i.i.d. assumption is violated (common violation in real data).

**Sufficient conditions for fast causal feature selection in large datasets.** Our causal discovery theorem imposes assumptions that are commonly met in real datasets where candidate variables have one known cause. We proved that our proposed conditions, under Assumptions 1-5, are sufficient for the identification of direct or indirect causes of a target variable. Thus, we can rule out that the measured dependency between the causal variable and the response is due to a confounding path, even due to a hidden variable. However, our procedure may not identify all causes (see Fig. 5 in suppl.). Simulations yielded successful application of our algorithm with very low percentages of false positive in dense and large graphs. The robustness of our algorithm against confounders,

alongside the linear scaling of complexity, render it suitable for causal feature selection in large datasets, where false acceptance is considered much more serious compared to false rejection.

**Not an instrumental variable approach.** Note that although our assumption about the existence of a path from $P$ to $M$ (Assump. 4) resembles part of the definition for instrumental variables (IV) [32, 33], it is not. To apply our method, in contrast to IV, we do not assume any independence of variable $P$ from unobserved variables that may affect $M$ and $R$ as hidden confounders, nor do we assume the lack of a directed path from $P$ to $R$ that does not include $M$ ("exclusion restriction"). In our setting, we do not assume that the variables $P^i$ are exogenous variables as in [34]. Note also that the approach of [35] is not applicable here because [35] assume that none of the other observed variables are descendants of the potential cause and the target variable. We don't have any prior knowledge of this kind, apart from the time order. Further, [35] need to search for a set of (possibly multiple) variables to condition on, raising the known statistical problems.

**Neurophysiological validity of results.** The application of our proposed method on our EEG data gave performance-specific causes across subjects, which are consistent with the known roles of physiological $\alpha$, $\beta$ and $\gamma$ brain rhythms in upper-limb movements. In particular, $\beta$ activity has been found significantly elevated in patients with motor disorders (tremors, slowed movements) such as Parkinson's disease [16, 36, 37]. Furthermore, in healthy subjects, elevated $\beta$-power has been found to play an antikinetic role [37]. Our findings support this conclusion, as we found channels in the $\beta$ power to play a causal role for subjects that did not improve their motor performance. On the other hand, increased $\gamma$ activity over the motor cortices has been associated with large ballistic movements [13, 12]. It has also been suggested to be prokinetic, given that it is increased during voluntary movement [38]. Our findings appear in accordance with this conclusion, since our method detected causal motor channels in the $\gamma$ band, in subjects who managed to reduce their reaching times and improved their motor performance. Moreover, our detected causal channels in the ipsilateral hemisphere at $\alpha$-band are consistent with neurophysiological studies that report increased $\alpha$-power over ipsilateral sensorimotor cortex during selection of movement [39]. Yet, no association of $\alpha$-power and motor performance has been reported. Although there is no ground truth for comparing our neurophysiological results, the findings appear at least plausible given current understanding of the aforementioned physiological brain rhythms in movement. Therefore, our method contributes to the more detailed localisation of causal cortical electrode-areas.

Finally, we want to emphasize on the appropriate way of interpreting our neurophysiological results. Since EEG electrodes record mixtures of underlying neuronal activity, and, therefore, are macro-variables, one could argue about their adequacy as variables for causal inference [40]. In order to consider EEG electrodes as appropriate causal candidates, we assume that the power measured on the electrode level mostly depicts the cortical activity right underneath. We can then interpret our causal findings as the brain activity which plays a causal role for the motor performance we observe. This detection of causal features sheds more light on the underlying cortical mechanism that acts during upper-limb movements. However, as it is still unknown how the stimulation current in a specific frequency interacts with ongoing brain oscillations, there is not a one-to-one mapping between the causal brain features and the stimulation targets. For example, as it has been shown in [17] $\beta$-rhythms may act as a mediator of $\gamma$ stimulation to motor performance. In the chain *stimulation parameters* $\rightarrow$ *brain activity* $\rightarrow$ *response*, our causal method contributes to the second link; thus it narrows the question of personalised stimulation to *stimulation parameters* $\rightarrow$ *detected causal brain activity*. Hence, the search for personalised stimulation parameters can be reduced to the detection of those that upper- or down-modulate accordingly the causal brain features which our algorithm identifies.

**Contribution.** We propose an algorithm and prove a theorem that allows to identify direct or indirect causes of a response variable, tailored to problems in which a cause of a candidate cause is known. This can naturally happen in set-ups where two nodes constitute consecutive time stamps of a variable's state in a system, and an edge from the previous to the present state can be assumed. The number of required CI tests is reduced to one targeted CI test per variable with one conditioning variable. Therefore, the complexity of the algorithm scales linearly with the number of variables. Furthermore, we thus need a substantially weaker version of faithfulness. We also show why our method is robust against violation of the i.i.d. assumption, assuming we can model the time effect as an independent variable. Finally, applying our algorithm on EEG data exhibited results with rigid consistency with current neuroscientific conclusions, helping to step closer towards personalized stimulation.

**Acknowledgments**

Authors would like to thank Sebastian Weichwald and Mateo Rojas-Carulla for their interesting feedback.

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
