[Supplementary Material]

# A  Supplementary material

## A.1  Definitions

We briefly introduce some fundamental definitions of Causal Bayesian Networks [20], which we use in our paper to present and prove our methodology. For a thorough study see [20, 21, 22].

**Definition** (d-separation [3]).  *In a directed acyclic graph (DAG) $G$, a path between nodes $I_1$ and $I_m$ is blocked by a set $\mathcal{S}$ (with neither $I_1$ nor $I_m$ in $\mathcal{S}$) whenever there is a node $I_k, k = 2, ..., m - 1$ , such that one of the following two possibilities holds:*

*(i)  $I_k \in \mathcal{S}$ and $I_{k-1} \rightarrow I_k \rightarrow I_{k+1}$ or $I_{k-1} \leftarrow I_k \leftarrow I_{k+1}$ or $I_{k-1} \leftarrow I_k \rightarrow I_{k+1}$*

*(ii)  Neither $I_k$ nor any of its descendants is in $\mathcal{S}$ and $I_{k-1} \rightarrow I_k \leftarrow I_{k+1}$.*

*In a DAG $G$, we say that two nodes $A$ and $B$ are d-separated by a third node $C$ if every path between nodes $A$ and $B$ is blocked by $C$. We then write $A \perp\!\!\!\perp_G B \mid C$.*

**Definition** (Ancestor, Descendant [22]).  *An ancestor of a node $A$ is any node $B$ such that there is a directed path from $B$ to $A$. A descendant of a node $A$ is any node $B$ such that there is a directed path from $A$ to $B$.*

**Definition** (Causal Markov Condition [22]).  *Let $G$ be a causal graph with vertex set $\mathcal{V}$ and $P$ be a probability distribution over the vertices in $V$ generated by the causal structure represented by $G$. $G$ and $P$ satisfy the Causal Markov Condition if and only if for every $W$ in $\mathcal{V}$, $W$ is independent of $\mathcal{V} \setminus (Descendants(W) \cup Parents(W))$ given $Parents(W)$.*

*Here we use the global version of Markov condition, which reads: if $\mathcal{X} \perp\!\!\!\perp_G \mathcal{Y} \mid \mathcal{Z} \Rightarrow \mathcal{X} \perp\!\!\!\perp \mathcal{Y} \mid \mathcal{Z}$ for all disjoint vertex sets $\mathcal{X}, \mathcal{Y}, \mathcal{Z}$ (where $\perp\!\!\!\perp_G$ denotes d-separation, as defined above)*

## A.2  Sufficient but not necessary

We present an example where both direct causes are rejected. In this example although both $M^1$ and $M^2$ nodes are causes of $R$, both are rejected. $P^1$ and $R$ are not d-separated by $M^1$ due to the path including $P^2$ and $M^2$ or because $M^1$ acts as a collider. On the other hand, $P^2$ and $R$ are not d-separated by $M^2$ due to the path including $P^1$ and $M^1$.

**Figure 5:** Example of DAG where causes are rejected because our theorem is sufficient but not necessary. Here, if all direct edges are equally strong, then both $M^1$ and $M^2$ are not identified by our theorem, due to the confounding path formed by the $P$ variables.

## A.3  Modelling of the i.i.d. assumption as a hidden time variable T

Here we present the graphs for the heuristic argument that we present in Section 4, suggesting that our method is robust in regard of i.i.d. violation of the random variables in the DAGs. To this end, we model the time dependence formally by a hidden time variable $T$. $T$ can affect the DAG according to the seven possible scenarios depicted in Figure 6 for the case that $M^i$ is a cause of $R$, and according to the seven possible scenarios depicted in Figure 7 in case $M^i$ is not a cause of $R$.

Graph (a) corresponds to the case that time dependence affects all observed variables of our DAG. Graph (b) describes the case where time dependence affects only the causal candidates of our DAGs. Graph (c) depicts the case that time dependence affects the previous state of candidate variables and the response variable. Graph (d) presents the case where the time dependence affects the current state

of the causal candidate variables and the response variable. Graphs (e), (f) and (g) depict the cases that time dependence affects only one variable of the DAG at a time.

As already described in Section 4, we examine each of these graphs with our theorem 2.2. In case that $M^i \dashrightarrow R$ exists, in the graphs (a), (c) and (d) the second condition of theorem 2.2 is violated, since $P^i \not\perp\!\!\!\perp R \mid M^i$. In all the other graphs of Figure 6, the cause variable $M^i$ is correctly identified.

**Figure 6:** Different possible effects of the trial variable $T$ on the DAG, in the case that the i.i.d. assumption is violated. Case I: $M^i \dashrightarrow R$ exists.

If $M^i \not\dashrightarrow R$ (figure 7), graphs (a), (c) and (d) comply with condition (1) of theorem 2.2, but violate condition (2), correctly rejecting the variable $M^i$. For the rest graphs the variables already violate condition (1) and so are rejected. Hence, if the hidden variable $T$ is present, some causal variables may be rejected by theorem 2.2 but no non-causal variable is falsely accepted.

**Figure 7:** Different possible effects of the trial variable $T$ on the DAG, in the case that the i.i.d. assumption is violated. Case II: $M^i \not\dashrightarrow R$.

## A.4 False positives and false negatives for different simulated DAGs

Here we present in two separate figures the false positives and false negatives described in Figure 2.

**Figure 8:** Percentage of false positives calculated on the number of $n$ candidate features, over 20 random simulated graphs, for different number of $i$ candidate features ($n = 5, 20, 50, 125, 200$), different Bernoulli probability to define sparsity and different number of samples (100, 200, 300, 400, 600, 800, 1000).

**Figure 9:** Percentage of false negatives calculated on the number of $n$ candidate features, over 20 random simulated graphs, for different number of $i$ candidate features ($n = 5, 20, 50, 125, 200$), different Bernoulli probability to define sparsity and different number of samples (100, 200, 300, 400, 600, 800, 1000).

In Figures 8 and 9 the existence of an edge between the nodes of our simulated graphs is defined by a Bernoulli distribution with probability $p = 0.2, 0.3, 0.4$ and $0.5$ respectively. Figure 8 depicts the percentage of false positives over twenty random graphs, for each combination of number of $M^i$ nodes $n$, samples and sparsity of edges. False positives occurring due to statistical error in the computation of the dependencies and conditional independences are very few, slightly increasing as the number of nodes increases, and with a tendency to reduce with more samples. Clearly, the probability of false positives increases with the number of nodes. The number of false negatives (Fig. 9) appears inflated because we consider as true causes both the direct and the indirect ones; so in case only the direct cause is correctly identified, then its ancestors who are indirect causes will be counted as false negatives. That is why the more dense are the edges of the graph, the more false negatives appear.

## A.5 Detected causal features for all subjects - Grouping and explanation of results based on reaching times

Here we present the detected causes for all subjects we processed with our causal method. In total, our algorithm detected causal brain features in seventeen out of twenty-one subjects. Our findings group subjects in three main categories that couple detected causes with subject's performance: those that $\gamma$ power is detected when subjects improve their performance (Figure 12), those that $\beta$ power is detected when subjects worsen or do not improve their performance (Figure 10), and finally those that $\alpha$ power is detected in the ipsilateral hemisphere (Figure 13).

**(a)**

**(b)**

**(c)**

**(d)**

**Figure 10:** Electrodes over contralateral motor cortex in the beta power at subjects that remain stable or worsen during the reaching trial, are detected as causal features from our algorithm. Y-axis is in logarithmic scale.

Figure 10 depicts the subjects that did not improve their movement duration throughout the sequence of reaching trials or who got worse (larger durations for completing the trial). We observe that our algorithm detects causes over motor channels in the beta range (second headplot), which is consistent with the literature findings about the predominant role of beta power in slow or unstable movements. In addition, we observe that among these subjects, for those who in general had performances better than the average, our algorithm detects also some electrodes in the gamma range (fourth headplot), which complies with the facilitatory role of gamma from the literature.

**(a)**

**(b)**

**(c)**

Alpha [8 12]Hz    Beta [12 25]Hz    Low gamma [25 45]Hz    Gamma [60 80]Hz

Subject: EE

Motor performance

**(d)**

Alpha [8 12]Hz    Beta [12 25]Hz    Low gamma [25 45]Hz    Gamma [60 80]Hz

Subject: CD

Motor performance

**(e)**

Alpha [8 12]Hz    Beta [12 25]Hz    Low gamma [25 45]Hz    Gamma [60 80]Hz

Subject: DD

Motor performance

**(f)**

**(g)**

**(a)**

**(b)**

**Figure 12:** Electrodes over contralateral motor cortex in the low and high gamma power at subjects that improve their reaching movement duration over the trials, are detected as causal features from our algorithm. Y-axis is in logarithmic scale.

Figure 12 depicts the subjects that improved their movement, decreasing the duration of their reaching movements throughout the sequence of the trials. We observe that our algorithm detects causes over motor channels in the gamma range (3rd and 4th headplot), which is in accordance with the facilitatory role of cortical gamma power in motor performance. For subjects whose average performance is far below the median performance despite their improvement, also some electrodes in beta range arise.

**(a)**

**(b)**

**Figure 13:** Electrodes mainly over ipsilateral motor cortex in the alpha power, are detected as causal features from our algorithm in some subjects. Y-axis is in logarithmic scale.

Figure 13 depicts the subjects for who our algorithm detected causes over ipsilateral motor channels in the alpha range (1st headplot). For subject JJ, who slightly improves her duration times in the end, gamma power also arises as causal feature on the contralateral motor cortex. Finally on subject HG channels on both contralateral and ipsilateral cortex were detected as causal. Our findings of causal channels in the ipsilateral motor cortex at $\alpha$-band are consistent with neurophysiological studies giving evidence of increased $\alpha$-band power over ipsilateral sensorimotor cortex during selection of movement [39]. Yet, no association of alpha power and motor performance has been reported.

**(a)**

**(b)**

**Figure 14:** For subjects AA and KL with very slight improvement of reaching movements either in the middle or in the end of the experiment, our algorithm detected one electrode at the gamma range. Y-axis is in logarithmic scale.

Finally in Figure 14, for subjects AA and KL with very slight improvement of reaching times, which they don't manage to maintain, our algorithm detected one electrode at the gamma range, which may imply again the facilitatory role of gamma, which however is not strong enough to result in improved times.

| Subject | Alpha | Beta | Low Gamma | Gamma | Above Group Average | Performance |
|---|---|---|---|---|---|---|
| AA | - | - | - | CPP4h | False | Inhibited but then improved |
| AB | - | FC2, CCP1h, CPP1h, CP6 | - | - | False | Full inhibition |
| BA | - | - | - | - | True | Full improvement |
| BB | - | CCP2h, CCP4h, CPP4h | CP1 | FCz, FC4, CCP3h, CPP3h | False | Full improvement |
| CC | - | CCP6h | - | - | True | Inhibited but then improved |
| CD | - | - | C5 | FCC1h | True | Full improvement |
| DC | FCC5h | CPP2h, CP5, CPz | C2, CCP2h | FC5, CCP2h | False | Inhibited but then improved |
| DD | - | - | FCC2h | - | True | Improvement but then inhibited |
| EE | - | - | C2, C5, CCP4h | FC5, CPP5h | True | Improvement but then inhibited |
| FF | - | FC1, FCz, C2, C3, CPP4h, FCC3h | FC3, FC1, C2, C4, C3, CCP4h, CPP6h, CPP1h, FCC5h, CP3 | FC1, CCP5h, CPP1h, FCC4h, FCC6h, CP5, CP4, CP6 | True | Improvement but then inhibited |
| GG | - | FC4 | - | FC4 | False | Inhibited but then improved |
| GH | - | - | - | - | True | Full improvement |
| HG | C2, C1, CCP3h, CPP5h | CP1 | - | - | True | Full improvement |
| HH | FC2, FCz | - | - | - | True | Improvement but then inhibited |
| II | - | - | - | FC2, FC4 | False | Full improvement |
| IJ | CP5 | - | FC3, FC6, C4 | FC5, FC3, FC4, FC6, C2, C4, C6, CCP2h, CCP4h, CPP6h, FCC5h, FCC2h, FCC6h, CP3 | False | Full improvement |
| JI | C2 | - | - | - | False | Full improvement |
| JJ | FC4, FC6 | - | - | FCC5h, CP1 | False | Full improvement |
| KK | C6, CP2 | CCP4h, CCP3h, FCC3h, FCC5h, FCC6h, CP3 | FCC2h, CP3, CP1, CP2 | FC5, FC6, C6, CCP4h, CCP6h, FCC3h | False | Full improvement |
| KL | - | - | - | C1 | False | Improvement but then inhibited |
| LL | - | FCC3h | - | FCC2h | False | Full improvement |

**Table 3:** Detected causal features for all twenty-one subjects.