[Reviews · NeurIPS 2019]

Reviewer 1



Summary: Conditional Independence Testing is an important part of causal structure learning algorithms. However, in the most general case either one has to do a lot of conditional independence tests and/or test by conditioning on a very large number of variables. This work proposes using at most two CI tests per candidate parent involving exactly at most one conditioning variable to filter the real parents of a response variable under certain conditions. This work is interested in identifying direct causes of a Response variable from amongst a set of a candidate parent variables {M_i}. Response variable does not have any observed descendants. Suppose, each candidate parent has a known parent P_i and suppose, there is no causal path from R to M's or from M variables to P variables, authors show that by doing two conditional independence tests per candidate parent, false positives can be made to be 0. That is if a perfect CI oracle is used the declared parents are always real under causal faithfulness. I checked the proofs. They are correct to the best of my knowledge. In fact, the results also hold under common latents between P and the M variables. Authors essentially say that through EEG cortical activity in certain areas of the brain can be observed during planning, doing action where action is movement of the limb. Response is the time it takes to move the limb within a certain duration. Special EEG data is collected where patients are asked to plan to move, then attempt to move within a certain duration. So the plan stage cortical activity would be the parent (P_i) of the cortical activity during movement (M_i) and it is the same variable across time. The authors argue that their data fits the model they propose and hence their tests have very low false positive rates. They conduct tests where a candidate brain feature is log power of the cortical activity in various frequency bands at various positions. Authors show that for patients that are not able to move limbs well, certain frequencies in certain areas become causal factors and in people with good motor actions, certain other frequencies become causal factors. They provide details of actual experiments done with patients and corroborate with neuroscience experiments. Originality: The notion of doing efficient CI tests with appropriate assumptions accompanied with the EEG experiment and final results corroborating with neuroscience literature makes it very original. Significance: Goes without saying that this would inspire neuroscience community to adopt such techniques directly inspired by CI testing and causality with DAG models. Further, it is a great work to follow up even within the causality community. Clarity: The experimental setup, the mapping to their model with P,M and R variables. Justification of using non-i.i.d data with time as latent variables and showing that it does not affect their conclusions are all very clear. Quality: Definitely very good quality empirics combined with some very cute theoretical observation.

Reviewer 2



The paper proposes a constraint-based method for selecting causal features for a given target variable in an neuroimaging context. Theoretical results are provided which result in an algorithm that requires only one independence test with one conditioning variable per possible causal feature and a weaker notion of faithfulness. Empirical results on simulated data show low false positive rates and relatively low false negatives for small to medium numbers of variables. Empirical results on real data area encouraging. In general the paper is well written though the section on empirical results on real data could be improved to be more clear. The method presented is novel and sound. This may be significant as causal discovery from neuroimaging data is currently a popular area of study. However, it is somewhat difficult to determine how strong these results are since they are not compared to any baseline. It would be helpful if the paper spoke more about the fact that the conditions are only sufficient, but not necessary and the likely degree to which it may fail to identify causes (either characterize this theoretically or show empirically). The false negative rates do not appear to converge in the simulations. It would be interesting to know if they do (eventually) if the samples are increased or if this is a result of the fact that the method fails to identify some causes and the degree to which this increases with n.

Reviewer 3



My main concern for the paper was the lack of comparison with other methods. The authors explained in their response that the reason was that the dataset was collected anew, and they are going to release the data. The conclusion drew from the analysis is in accordance with the general consensus on the neuroscience problem, which serves as a weak validation of the method. One reviewer does point out that the authors made some efforts to compare their methods with existing Lasso variants on synthetic datasets. Thus I am inclined to raise my score to a weak accept It will be great if the authors can be more clear on the advantages of their methods, comparing to other methods. In my opinion, I think it is better explained in the response than in the paper. ---------------------Before Rebuttal--------- This paper presents a series of simplifying assumptions for causal inference and proves that only a single conditional independence test per candidate cause is required in such a regime. However, it is not very clear how the assumptions required for the algorithms are relaxed compared to existing algorithms -- only causal sufficiency is removed and only the PC method requires this assumption. The major drawbacks of this paper are due to the lack of comparison with existing methods. It’s not very clear whether under the constrained conditions if other methods are applicable. In Figure 2, it appears that the large number of false negatives are okay, however this would only make sense if the method was only designed to capture direct causes. However, the method should capture both. It would be nice to see an explanation of why these false negatives appear. Figure 1 is rather small, but important for understanding the type of DAGs assumed by this method. It should be made larger.

[Author Response · NeurIPS 2019]

Many thanks for the helpful comments. We appreciate the praise for the "Extremely good and unique empirical contributions", "Very nice theoretical results" and "novel and sound" in the two high-confidence reviews recommending acceptance, as well as the constructive criticisms, which we will address below.

**Markov blanket, Lasso and comparison with other methods:** We tried LASSO prior to this work, but the results were not neurophysiologically meaningful. This is understandable in retrospect since causal feature selection via Lasso or Markov Blanket (MB) requires causal sufficiency, let alone curse of dimensionality. Furthermore, with high dimensional data, any algorithm using CI tests has to condition on large variable sets, in which case CI testing is hard and cannot be trusted unless sample sizes are huge. Finally, even if causal sufficiency were to hold, the known MB detection algorithms and Lasso do not *detect* variables but rank them, and gradually evaluate the prediction accuracy by including more variables, according to the ranked order the algorithm returned. This requires a heuristic *hyperparameter* to define what is the right acceptable number of variables to be included in the MB, which both affects the FP and FN and does not provide a straight forward metric (FP, FN) to compare with our method. For completeness, however, below we provide comparison results (to be included in the final version) of our method against available algorithms (average for 10 random graphs): HSIC Lasso (Yamada, 2014), Backwards elimination (BE) with HSIC, and Forward selection (FS) with HSIC for MB discovery (Song, 2007). Lacking space, we selected to examine the most optimistic for the other algorithms case, that of large sample size (800) and two cases of small (20) and large graphs (125 nodes), for sparse (0.2) and dense (0.5, more true causes) edges. We report the % of FP and FN in the number of variables. In sparse large graphs FS gives more FP. Lasso and FS give more FP in small sparse and dense graphs. BE performs worse in small sparse graphs. Overall, our method manages to keep FPs very low ($\sim 2.1\%$) for all dense/sparse, small/large graphs, while other algos' performance varies with the case. Optimal parameters based on the true number of causes was selected for Lasso. BE and FS computations took significantly long. Furthermore, we stress that in these simulations no hidden variables exist, which is an extra advantage for the compared algorithms.

| (nodes, sparse) | (20,.2) | (20,.5) | (125,.2) | (125,.5) | (20,.2) | (20,.5) | (125,.2) | (125,.5) | (20,.2) | (20,.5) | (125,.2) | (125,.5) | (20,.2) | (20,.5) | (125,.2) | (125,.5) |
|---|---|---|---|---|---|---|---|---|---|---|---|---|---|---|---|---|
| | Our method | | | | Hsic Lasso | | | | BE hsic | | | | FS hsic | | | |
| FP(%) /FN(%) | **3.5**/31.5 | **2**/80 | **2.9**/70.3 | **0**/80.8 | 9.5/22.5 | 5.5/47.5 | 1.1/77.4 | 0/84.8 | 11/23 | 1.5/79 | 1.4/77.9 | 0/97.6 | 6/25 | 7.5/26 | 7.8/47.4 | 1.1/14.5 |

**Data will be made public /lack of ground truth:** The reason why there is no baseline comparison for our EEG results is that this EEG data have not been used for causal inference analysis before, as we ourselves recorded it for this paper. We will indeed make it public upon acceptance, to contribute this fascinating dataset to inspire further causality research in this field that depends on it. Lack of ground truth is a common problem in brain datasets, which is why we compare to conclusions and findings in the literature. Our findings are in accordance with established neuroscientific conclusions about the brain rhythms present during movement in different conditions and we thus believe they are meaningful.

**Sufficient but not necessary:** The fact that our conditions are sufficient but not necessary may potentially lead to fewer detected causes. Indeed, an empirical example of this was given in section A2, fig. 5 of suppl. submitted alongside the paper, showing a case where both direct causes $M_1$ and $M_2$ of target $R$ are rejected. In this example, $P_1$ and $R$ are not d-separated by $M_1$ because $M_1$ is a collider. Moreover, $P_2$ and $R$ are not d-separated by $M_2$ due to the path including $P_1$ and $M_1$. In this example there are instantaneous effects between the $P$ and the $M$ stages of the causes. This leads to rejecting both causes. This is a counterexample where although the variables are causes of the target, our conditions are not met; thus the ($\leftarrow$) direction of our theorem cannot be proved, thus sufficient but not necessary.

**Further explanation on the False Negatives fig. 2:** Our method detects direct and indirect causes. If for example $A \rightarrow B \rightarrow C \rightarrow D$ and $E \rightarrow D$ in the same graph, our method could identify as causes of $D$ for instance $E$ (direct) and $A$ (indirect). In that case, $B$ and $C$ will be counted as FN, because they were not identified. However, in reality, this is not a problem, because we correctly identified $A$ which is a cause of $D$ (as well as of $B$ and $C$), and so if we intervene on $A$ we will affect $D$, which is our ultimate purpose, supposing i.e. $A, B, C, E$ are brain regions and $D$ is the arm speed. Therefore, the number of FN (suppl. fig.9) appears inflated because we consider as causes both the direct and the indirect ones. In case only the direct cause is identified, then its ancestors (indirect causes) will be counted as FN. That is why the number of FN increases with the number of features $n$ and the density of the graph. We stress that the reason why we care more about the FP, is because we address causal problems where a false rejection is less harmful than a false acceptance. If a brain area is falsely identified as a target for stimulation it can be harmful, compared to the harmless case that not all areas are identified. There is no free lunch, and this is a small price to pay to get the linear computation time and the statistical significance of our CI tests with one targeted conditioning variable.

**Improvements upon previous methods:** 1. To the best of our knowledge, this is the first constraint based algorithm that scales linearly with the number of variables. Previous methods based on CI tests grow exponentially in time with the number of variables, (if sparse data then they grow polynomially), as they require more than one CI test per variable. Therefore, we greatly reduce the computational complexity. 2. Our algorithm builds on tests that condition on only one variable each; previous methods require conditioning on many variables. With this improvement, the statistical strength of our inference is superior compared to cases where there is more than one conditioning variable. Furthermore, due to this improvement, as *reviewer #2* also pointed out, we require a weaker notion of faithfulness. 3. Our method does not assume causal sufficiency - a common assumption which is, however, often violated in real datasets. 4. Finally, although originally for completeness we assume i.i.d. samples, we prove in the suppl. that our method is robust against false positives when the i.i.d. assumption is violated (common violation in real data).

[Meta-Review · NeurIPS 2019]

In this paper, the authors describe a novel causal discovery method that performs a single conditional independence test per features, and is thus scalable to high dimensional data, along with a novel encelographic data application. The reviewers appreciated the novelty of the methods, and the chosen application.